# Fast spin exchange across a multielectron mediator

Filip K. Malinowski[1], Frederico Martins [1], Thomas B. Smith [2], Stephen D. Bartlett[2], Andrew C. Doherty[2], Peter D. Nissen [1], Saeed Fallahi[3], Geoffrey C. Gardner[3], Michael J. Manfra [3,4], Charles M. Marcus [1] & Ferdinand Kuemmeth[1]

Scalable quantum processors require tunable two-qubit gates that are fast, coherent and long-range. The Heisenberg exchange interaction offers fast and coherent couplings for spin qubits, but is intrinsically short-ranged. Here, we demonstrate that its range can be increased by employing a multielectron quantum dot as a mediator, while preserving speed and coherence of the resulting spin-spin coupling. We do this by placing a large quantum dot with 50–100 electrons between a pair of two-electron double quantum dots that can be operated and measured simultaneously. Two-spin correlations identify coherent spin-exchange processes across the multielectron quantum dot. We further show that different physical regimes of the mediated exchange interaction allow a reduced susceptibility to charge noise at sweet spots, as well as positive and negative coupling strengths up to several gigahertz. These properties make multielectron dots attractive as scalable, voltage-controlled coherent coupling elements.

[1] Center for Quantum Devices and Station Q Copenhagen, Niels Bohr Institute, University of Copenhagen, 2100 Copenhagen, Denmark. [2] Centre for Engineered Quantum Systems, School of Physics, The University of Sydney, Sydney, NSW 2006, Australia. [3] Department of Physics and Astronomy, Station Q Purdue, and Birck Nanotechnology Center, Purdue University, West Lafayette, IN 47907, USA. [4] School of Electrical and Computer Engineering, and School of Materials Engineering, Purdue University, West Lafayette, IN 47907, USA. These authors contributed equally: Filip K. Malinowski, Frederico Martins. Correspondence and requests for materials should be addressed to F.K. (email: kuemmeth@nbi.dk)

The Heisenberg exchange interaction between neighboring quantum dots allows precise voltage control over spin dynamics, due to the ability to precisely control the overlap of orbital wavefunctions by gate electrodes. This allows the study of fundamental electronic phenomena[1–4] and finds applications in quantum information processing[5]. Although spin-based quantum circuits based on short-range exchange interactions are possible[6,7], the development of scalable, longer-range coupling schemes constitutes a critical challenge within the spin–qubit community. Approaches based on capacitive coupling[8,9] and cavity-mediated interactions[10–12] effectively couple spin qubits[13,14] to the charge degree of freedom[15,16], making them susceptible to electrically-induced decoherence. The alternative is to extend the range of the Heisenberg exchange interaction by means of a quantum mediator[17–20].

Here, we show that a multielectron quantum dot with 50–100 electrons serves as an excellent mediator, preserving speed and coherence of the resulting spin–spin coupling while providing several functionalities that are of practical importance. These include speed (mediated two-qubit rates up to several gigahertz), distance (of order of a micrometer), voltage control, possibility of sweet spot operation[21,22] (reducing susceptibility to charge noise), and reversal of the interaction sign (useful for dynamical decoupling from noise)[4,23,24].

## Results

**Implementation of long-range coupling**. We implement long-range exchange coupling mediated by a multielectron quantum dot in a linear array of five quantum dots, as shown in Fig. 1a. The quintuple dot is defined in a GaAs two-dimensional electron gas by means of electrostatic gate electrodes deposited on top of the heterostructure (see Methods section). The middle dot is populated by a large even number of electrons, between 50 and 100 as estimated from the lithographic size of the device and the density of the two-dimensional electron gas. Its ground state is chosen to be spinless as described in ref. [4]. Two two-electron double dots are tunnel-coupled[4] on opposing sides of the large middle dot and are each initialized and read out using standard techniques for singlet–triplet qubits[5,25].

The exchange interaction is induced by a sequence of submicrosecond voltage pulses applied to the blue-colored gates in Fig. 1a, realizing the following steps (cf. Fig. 1b and further details about the pulse sequence in Supplementary Note 1). First, the outer dots are each populated by a pair of electrons. This initializes each double dot in the spin singlet state, $|S^{L/R}\rangle = (|\uparrow\downarrow\rangle - |\downarrow\uparrow\rangle)/\sqrt{2}$, where arrows indicate the spin state of the two electrons and the superscript L/R indicates the left and right double dot. Then, the electron pairs are rapidly separated within each double dot. Since electron wave functions effectively stop overlapping, this pulse turns off the exchange interaction within each double dot, allowing the outer dots to each store one reference spin. In the third step, $V_M$ is temporarily increased by $\varepsilon_M$, while negative (compensation) pulses are applied to all other gates (see Supplementary Note 1). This induces an exchange interaction between the inner one-electron dots mediated by the large dot. After the interaction time $\tau$ the exchange-inducing pulse is switched off. Subsequently, spin-to-charge conversion is used to read out the relative spin alignment within each double dot[5], independently and with single-shot fidelity (see Methods section).

The result of such a spin-exchange pulse sequence is shown in Fig. 2a. In the two panels, we plot the fraction of singlet outcomes ($P_S$) for each double dot as a function of $\tau$ and $\varepsilon_M$. Oscillations in $P_S$ witness exchange-driven flip-flop processes between the two spins located on the inner quantum dots. The oscillation

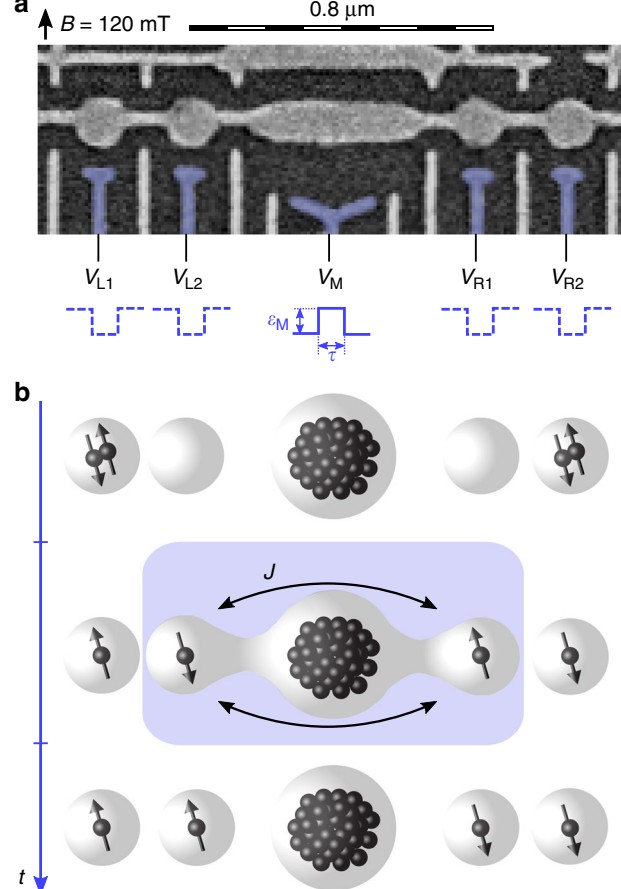

**Fig. 1** Detection of spin-exchange processes across a multielectron dot. **a** Scanning electron micrograph of the measured device. A multielectron dot is induced below the long segment of the horizontal gate electrode, while two two-electron double quantum dots are induced below its circular sections. Nanosecond voltage pulses applied to the blue-colored gates $V_j$ control the position of individual electrons and their mutual interactions. An external magnetic field (arrow) is applied in-plane of the device. **b** Operation steps. First, each double dot is initialized in a singlet state $|S^{L/R}\rangle$, by populating the outer dots with two electrons each. Then, single electrons are moved to the inner dots, thereby turning off their exchange interaction with the outer electrons, which serve as reference spins. Next, the exchange coupling $J$ between the inner electrons is induced, by temporarily raising $V_M$ by an amplitude parameterized by $\varepsilon_M$ (and lowering other gates to maintain constant overall charge). The exchange interaction causes flip-flops between electronic spins on the inner dots (for mechanisms, see Fig. 3a), which entangles the spin state of the left double dot with that of the right double dot. After an interaction time $\tau$ the resulting correlations in the relative alignment between inner spins and reference spins are detected by spin-to-charge conversion within each double dot, using two nearby sensor quantum dots (not shown)

frequency increases for larger values of $\varepsilon_M$. Consistent with complementary spin-leakage spectroscopy (see Supplementary Note 2), this indicates that positive pulses on gate $V_M$ lower the multielectron dot levels towards resonance with the inner dots, thereby increasing the rate of spin-exchange processes mediated by the multielectron dot.

**Correlated measurements**. Next, we demonstrate correlations between measurement outcomes for the left and right double dot. For fixed interaction time $\tau = 2$ ns, the demodulated voltage signals for the left and right sensor ($V_{rf,L}$ and $V_{rf,R}$)

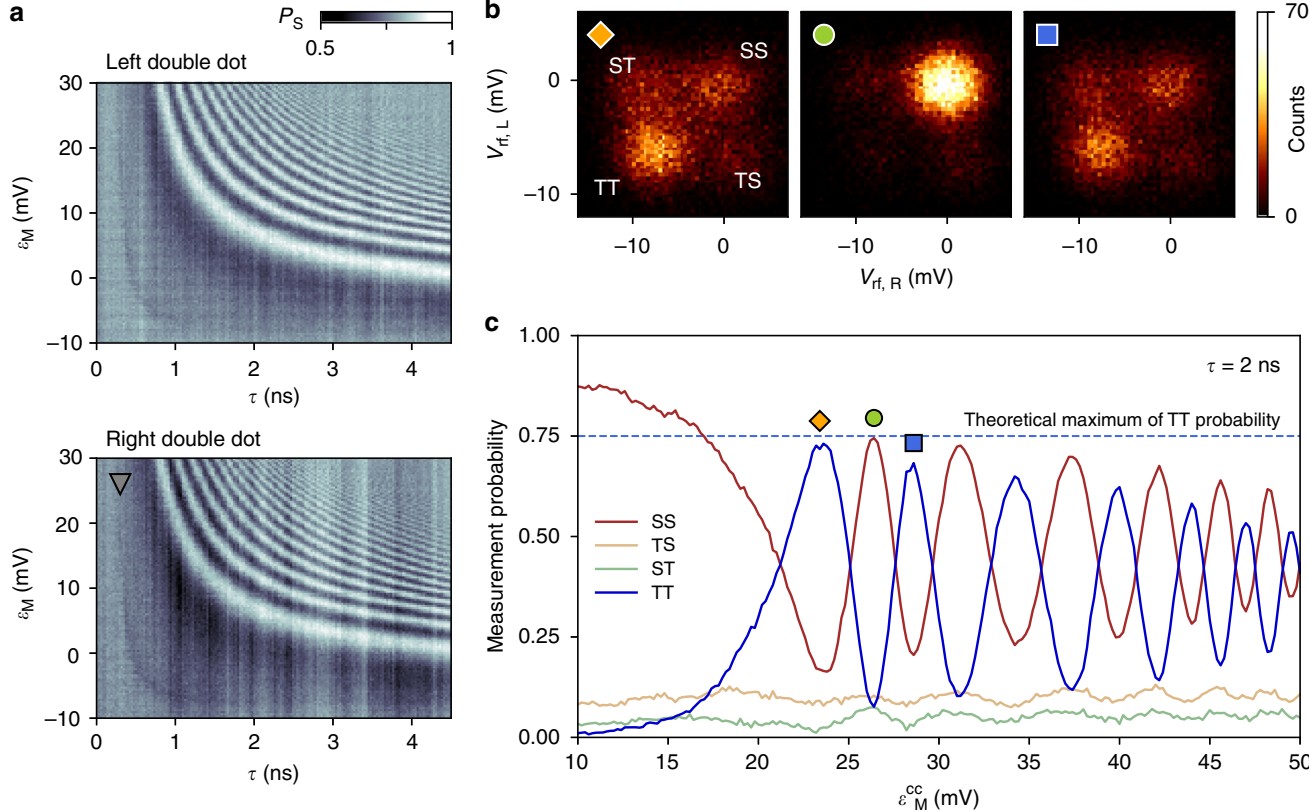

**Fig. 2** Exchange oscillations across the mediator and non-local correlations. **a** Fraction of detected singlet outcomes, $P_S$, acquired simultaneously for the left and right double dot, as a function of interaction time $\tau$ and pulse amplitude $\varepsilon_M$. The choice of detuning between inner dots, $\varepsilon = -2$ mV, corresponds to a symmetric operation point (cf. gray triangle in Fig. 3b). **b** Histograms of demodulated sensor voltages, when repeating a pulse cycle with $\tau = 2$ ns many times, for three different choices of $\varepsilon_M^{cc}$ as marked in panel (**c**). Counts bunch into four groups, each associated with a different combination of singlet (S) or triplet (T) measurement outcomes for the two double dots. Correlations within these single-shot measurement outcomes reveal the non-local nature of the interaction. **c** Joint probabilities of all four possible joint outcomes, as a function of the exchange-inducing pulse amplitude $\varepsilon_M^{cc}$ for fixed interaction time $\tau = 2$ ns. Here, $\varepsilon_M^{cc}$-pulses are defined similarly to $\varepsilon_M$-pulses, but with a different choice for the cross-compensation amplitudes (see Supplementary Note 1). Dashed line indicates the largest expected probability to detect TT for the maximally entangled state (see text)

exhibit correlations that oscillate with the amplitude of the applied pulse (Fig. 2b), confirming the non-local mechanism underlying panel 2a (in Fig. 2b, the exchange-inducing pulses parametrized by $\varepsilon_M^{cc}$ are defined similarly to $\varepsilon_M$-pulses in Fig. 2a, but employ more sophisticated cross-compensation pulses, as described in Supplementary Note 1). From these histograms, we extract the joint probabilities of detecting a singlet (S) or triplet (T) for the two double dots as a function of $\varepsilon_M^{cc}$ (see Supplementary Movie 1 for animated histograms). Figure 2c clearly shows anticorrelated probabilities for detecting SS and TT, whereas the probabilities of ST and TS are small and nearly constant. The joint probabilities were extracted by fitting the histograms with four Gaussians and correcting for double-dot relaxation during the measurement pulse (see Supplementary Note 3).

The oscillatory behavior of joint probabilities results from the precession between the initialized state $|S^L\rangle|S^R\rangle$ and the maximally entangled state $\frac{1}{2}(|S^L\rangle|S^R\rangle - |T_0^L\rangle|T_0^R\rangle + |T_+^L\rangle|T_-^R\rangle + |T_-^L\rangle|T_+^R\rangle)$. Here, the two kets denote the state of the left and right double dot, respectively, and the spin triplet states are labeled according to the standard convention, $(|T_0\rangle = (|\uparrow\downarrow\rangle + |\downarrow\uparrow\rangle)/\sqrt{2}, |T_+\rangle = |\uparrow\uparrow\rangle, |T_-\rangle = |\downarrow\downarrow\rangle)$. The coefficients associated with this entangled state explain the visibility in our measurement basis. For example, the maximum expected probability for TT is 75% in the case of perfect readout, nearly matched by the observed maxima in Fig. 2c. The observed

visibility for SS and TT is further reduced by residual counts of ST and TS. We attribute this background to unintentional dynamics of the reference spins in the outer dots, arising from decoherence due to their coupling to the nuclear spin bath associated with GaAs[26], and from the finite rise time of the voltage pulses.

In this experiment, we can exclude capacitive coupling as a mechanism leading to the observed oscillations, as it is small and manifests itself as a controlled phase gate between singlet–triplet qubits[8,9], not as a SWAP gate between single spins.

**Physical regimes of exchange interaction.** In Fig. 3, we identify different mechanisms of the exchange interaction mediated by the multielectron quantum dot. For that purpose, we define a new gate voltage parameter, $\varepsilon = (V_{L2} - V_{R1})/\sqrt{2} + C$ (where $C$ is a constant; see Supplementary Note 1), which controls the relative detuning between the two inner dots, and plot the readout probabilities $P_S(\varepsilon, \varepsilon_M)$. Since $\tau = 6$ ns is fixed for these measurements, each fringe in Fig. 3b corresponds to points of equal exchange energy $J$, while the density of fringes represents the gradient of $J$ (see Supplementary Note 8 for a discussion of finite-rise-time effects). The observed exchange strength rapidly increases when $\varepsilon_M > 20$ mV, especially for $\varepsilon \approx 0$, resulting in a high density of fringes that is blurred by a combination of aliasing and decoherence. For finite $|\varepsilon|$, the exchange increases more slowly, resulting in a pattern that is approximately symmetric with respect to $\varepsilon$. While Fig. 3 presents measurements

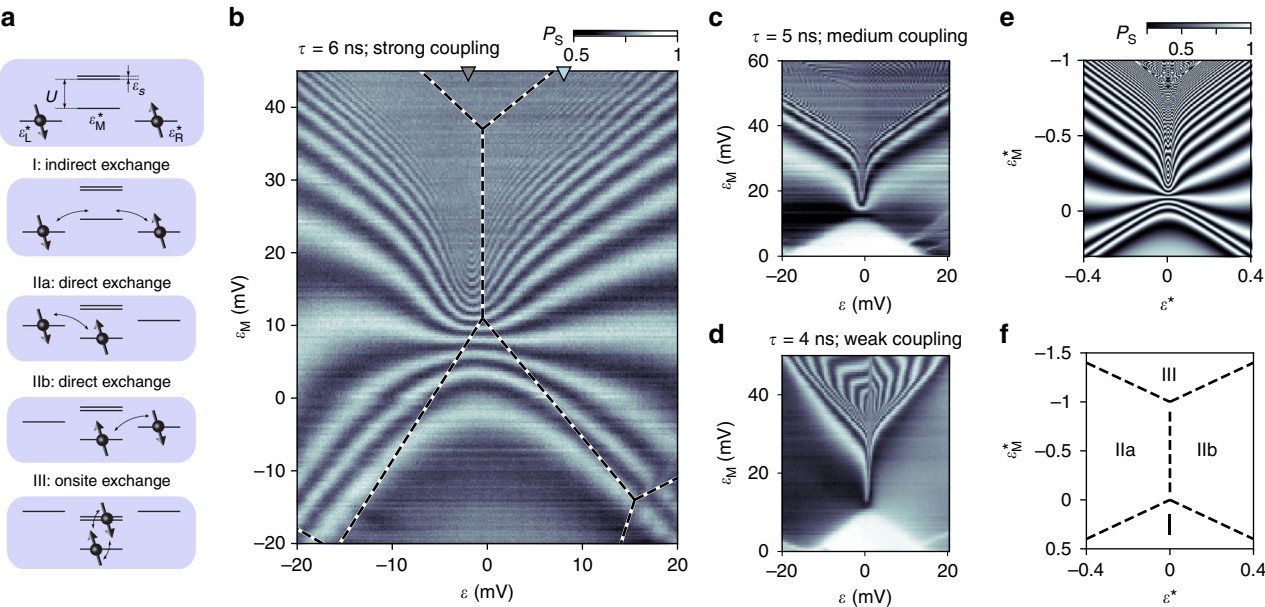

**Fig. 3** Physical regimes of exchange interaction. **a** In the Hubbard model, different spin-exchange processes dominate depending on the relative alignment of various single-particle levels (cf. Supplementary Note 7). Specifically, $\varepsilon_M^*$ is the single-particle energy of the lowest unoccupied orbital in the multielectron dot relative to the left and right orbital, $\varepsilon^* = (\varepsilon_L^* - \varepsilon_R^*)/2$ determines the relative detuning between the left and right orbital, and $U$ and $\varepsilon_S$ indicate the charging energy and the level spacing of the multielectron dot. Depending on which processes are energetically allowed or suppressed, we classify different regimes as illustrated. **b** Measured $P_S(\varepsilon, \varepsilon_M)$ for the right double dot for fixed interaction time $\tau = 6$ ns. Colored triangles indicate the detuning points used for Figs. 2a and 4b. Dashed lines indicate the location of independently measured charge transitions (see Supplementary Note 4). **c**, **d** Same as (**b**), but for reduced tunnel coupling between the multielectron dot and the inner dots. The interaction time is fixed at $\tau = 5$ and 4 ns in (**c**) and (**d**), respectively. **e** Simulated $P_S(\varepsilon^*, \varepsilon_M^*)$ in the Hubbard model. **f** Location of charge transitions (dashed lines) in the Hubbard model for the parameters used in (**e**). The corresponding charge configurations of the four regimes of exchange interaction are schematically indicated by dots in (**a**)

obtained from the right double dot, identical patterns are observed from the left double dot (see Supplementary Note 5). This is not the case when the occupancy of the multielectron dot is increased by one electron, as discussed in Supplementary Note 6.

The observed pattern can be understood by monitoring the charge distribution during the interaction step (Supplementary Note 4), and overlaying the fringes in Fig. 3b with the observed charge transitions (dashed lines). Consistent with simulations from the Hubbard model discussed below, shown in Fig. 3e, f, we identify each region with a different electron configuration, as illustrated by dots in Fig. 3a. In region I, the inner dots remain singly occupied, and the multielectron dot keeps its initial charge state. This corresponds to an indirect exchange interaction, where virtual tunneling through the multielectron quantum dot mediates superexchange[19]. In regions IIa and IIb, one of the electrons has relocated onto the multielectron dot, forming an effective spin-1/2 many-body state which directly exchange-couples to the other electron spin. The mirror symmetry of IIa and IIb with respect to $\varepsilon^* = 0$ reflects the left–right symmetry of the device, with minor deviations in the experimental data arising from a slight inequality in the tunneling barriers between the multielectron dot and the inner dots. In region III, the chemical potential of the multielectron dot is sufficiently low such that both electrons relocate onto the multielectron dot. Depending on their relative spin alignment, singlet-like or triplet-like, both electrons occupy either the lowest orbital, or the lowest and second lowest orbital, respectively. The energy difference between these spin configurations sets the coupling strength of this (rapid) onsite exchange interaction. It is related to two mesoscopic parameters, namely the single-particle level spacing of the two orbitals, and the spin correlation energy[4,27,28].

The different mechanisms of the exchange interaction can also be distinguished by reducing the tunnel couplings between the

multielectron and the inner dots (Fig. 3c, d). We observe $P_S \approx 1$ throughout region I, indicating that the reduced coupling effectively turns off the indirect exchange regime. Region I now appears separated by a sharp boundary, i.e., charge transition, from the direct exchange regions IIa,b. Similarly, in Fig. 3c, a sharp boundary between clear and blurred oscillation fringes is observed, indicating the transition between direct and onsite exchange regimes. Upon further reduction of tunnel coupling (Fig. 3d), an arc pattern is observed in region III. This behavior results from a sweet spot and a sign reversal of the exchange interaction[4,23], as argued below.

**Model and sweet-spot behavior.** To verify all four regimes, we evaluate a Hubbard model of the two inner quantum dots coupled to the multielectron quantum dot (see level structure in Fig. 3a and Supplementary Note 7). Using realistic parameters, this model qualitatively reproduces data in Fig. 3b, including the fringe pattern and the charge stability diagram (Fig. 3e, f).

Quantitative insight into the fast dynamics of onsite exchange can be obtained by reducing $\tau$ to 2 ns. This circumvents blurring and aliasing effects, revealing a characteristic arc pattern at the transition between direct and onsite exchange regimes (Fig. 4a). This pattern (similar to the one observed in Fig. 3d) is in fact a fingerprint of the exchange profile $J(\varepsilon, \varepsilon_M)$[22]: Retaking any pixel, say along a cut at $\varepsilon = -8$ mV (blue triangle in Fig. 3b), as a function of $\tau$ results in oscillations with frequency $f = J/h$[21]. Extracting $f$ for the cut shown in Fig. 4b reveals a non-monotonic behavior of the exchange coupling with respect to $\varepsilon_M$ (Fig. 4d). The presence of a maximum in frequency followed by a zero crossing is similar to exchange profiles studied in refs. [4,23], and arises if direct exchange (which depends on orbital-specific tunnel matrix elements) competes with onsite exchange (which depends on

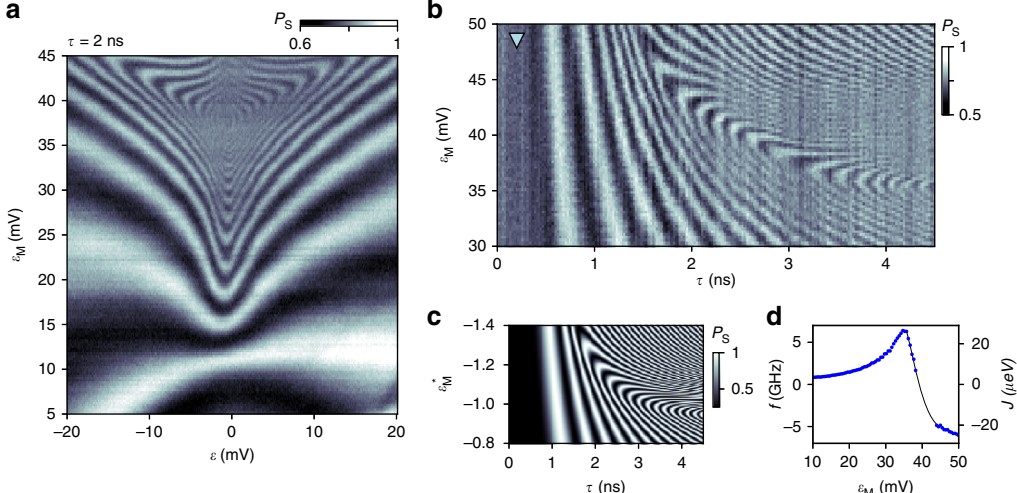

**Fig. 4** Sweet-spot behavior and competition between direct and onsite processes. **a** $P_S(\varepsilon, \varepsilon_M)$ for reduced interaction time $\tau = 2$ ns. A fingerprint pattern, related to a sweet spot in the exchange profile $J(\varepsilon, \varepsilon_M)$, emerges at the crossover from direct to onsite regimes. Measured (**b**) and simulated (**c**) time-dependent exchange oscillations for fixed $\varepsilon = 8$ mV (marked by blue triangle in Fig. 3b). Enhanced oscillation visibility along the chevron pattern indicates that operation at the sweet spot prolongs coherence. **d** Coupling strength of the quantum mediator, demonstrating high speed (GHz), sweet spot (maximum), and sign reversal, controlled by small voltage changes in $\varepsilon_M$. Data points represent the oscillation frequency extracted from rows in panel (**b**), which we identify with the exchange coupling strength $J = hf$. Solid line is a guide to the eye

electron correlation effects and, for relatively small orbital level spacing, can be negative). Accordingly, to qualitatively reproduce the chevron pattern of Fig. 4b, we must include two unoccupied orbitals of the multielectron quantum dot[4,24], as well as a finite rise time of the applied voltage pulses (see Methods and Supplementary Note 7). The results of such a simulation are shown in Fig. 4d and are in good agreement with experimental data.

Furthermore, the observed visibility of oscillations in Fig. 4b depends on $\varepsilon_M$, which we associate with an enhancement of fidelity in two operating regimes. First, for large values of $\varepsilon_M$, the onsite exchange energy is set by the (mesoscopic) level spacing of the dot, which to lowest order is insensitive to pulse amplitudes. This regime is akin to the noise-insensitive regimes noted in refs. [14,29] and exploited by the three-electron double-dot hybrid qubit[30,31]. Second, high-fidelity oscillations appear along the curved chevron pattern, suggesting that the local extremum in the exchange strength provides insensitivity[21,22] to fluctuations in $\varepsilon_M$. For this tuning of the device, the oscillation frequency in both noise-insensitive regimes exceeds 5 GHz, making it challenging to perform small angle rotations using conventional pulse generators. However, by decreasing the tunnel couplings between the multielectron dot and the inner dots, the operating speed at the sweet spot can be reduced as desired (down to 1 GHz as demonstrated in ref. [23]).

## Discussion
The significant coupling strength suggests realistic application in quantum information processing where fast spin exchange is needed between non-nearest neighbors. An alternative approach for coupling distant spin qubits (based on superconducting cavities and spin–photon coupling) is expected to be orders of magnitude slower, although no photon-mediated spin–spin coupling has been demonstrated to date (for comparison, strong spin–photon exchange itself has been observed, and is rather slow[10,11]).

Phenomenologically, the most striking difference between indirect exchange in our device and similar manifestations in a doubly-occupied triple dot[17] is its occurrence in a charge configuration that is left–right symmetric (see Supplementary Note 9). During this indirect exchange, two singlet-correlated

electron pairs are coherently linked over a linear array of five dots, making this the largest coherent quantum dot array to date. Microscopically, we speculate that the underlying mechanism can be viewed as a small-system manifestation of the Ruderman–Kittel–Kasuya–Yosida (RKKY) interaction, which has no such counterpart in the direct and onsite regimes[19].

An interesting next step building upon this demonstration is to employ a multielectron quantum dot of larger dimensions, with multiple single-electron quantum dots around its perimeter. This will allow coherent coupling of arbitrary pairs of electrons, and may lead to a programmable hardware architecture in which qubit–qubit connectivities can be reconfigured in situ to best serve the specific computational tasks. Increasing the coupler size has additional advantages, such as reducing the onsite exchange energy which would enable performing high-fidelity, small-angle rotations. Another direction is the implementation of this coupling scheme in silicon nanostructures, mitigating decoherence effects arising from the nuclear spin bath. Our demonstration of coherently swapping spin pairs across the multielectron quantum dot suggests that shuttling of individual electrons[32] through the multielectron quantum dot should also be feasible. Combinations of these achievements will open many paths for scaling quantum-dot-based qubit circuits.

## Methods
**Sample preparation**. The array of quantum dots is defined in a high-mobility (230 m$^2$ V$^{-1}$ s$^{-1}$ at 0.3 Kelvin) two-dimensional electron gas (density $2.5 \times 10^{15}$ m$^{-2}$) located 57 nm below the surface of a GaAs/AlGaAs heterostructure, by means of electrostatic gate electrodes deposited on top of the heterostructure[4]. A 10-nm-thick layer of HfO$_2$ is deposited on top of the heterostructure, prior to patterning the gold electrodes by electron-beam and lift-off lithography. A top view of the gate electrodes is shown in Fig. 1a. A detailed characterization of this device chip can be found in ref. [4].

**Readout**. Within each double dot, spin-to-charge conversion is used to read out the relative spin alignment within each double dot[5]. This is done at the end of each waveform cycle (cf. Supplementary Figure 1 in Supplementary Note 1). Specifically, a frequency-multiplexed measurement pulse reflected off two proximal radio-frequency quantum-dot-based charge sensors[25] allows us to distinguish between singlet and triplet states of each double dot, independently and with single-shot fidelity. To reduce errors arising from slow drifts in the demodulated sensor voltages ($V_{rf,L}, V_{rf,R}$), within each waveform cycle we acquire sensor voltages just after initialization of the singlet-singlet state ($V_{rf,L}^i$ and $V_{rf,R}^i$) and after the actual

interaction step ($V_{\mathrm{rf,L}}^{f}$ and $V_{\mathrm{rf,R}}^{f}$). These two measurements correspond to the "Reference measurement" and "measurement" segment indicated in Supplementary Figure 1. By defining the measurement outcomes relative to the reference outcomes, the adverse effects of slowly drifting sensor signals are effectively removed: $V_{\mathrm{rf,L}} \equiv V_{\mathrm{rf,L}}^{f} - V_{\mathrm{rf,L}}^{i}$ and $V_{\mathrm{rf,R}} \equiv V_{\mathrm{rf,R}}^{f} - V_{\mathrm{rf,R}}^{i}$.

**Implementation of exchange pulses with subnanosecond resolution.** To achieve subnanosecond resolution of the exchange pulse, we interfered two nominally canceling signals generated by two arbitrary waveform generator channels, and applied the combined signal to the multielectron-dot plunger gate $V_{\mathrm{M}}$. Specifically, we set the two channels to output a square waveform of identical duration and amplitude, but with opposite polarity, and combine them using an inverted power splitter. The pulse period is set to the repetition time of the intended pulse sequence, the rising edge of the pulse is set to the beginning of the intended exchange pulse, while the falling edge happens at the beginning of the double-dot initialization step. By finely adjusting the channel skew of the arbitrary waveform generator, positive or negative $V_{\mathrm{M}}$ pulses can be generated with subnanosecond control. While this method allows to overcome the limitations of the waveform generator's temporal resolution of 1.2 GS s$^{-1}$ (Tektronix AWG 5014C), the effective voltage pulse reaching the gate electrodes is still constrained by the 0.8 ns pulse rise time in our dilution refrigerator, resulting in distortion effects in Figs. 2–4 (cf. Supplementary Note 8).

### Data availability
The datasets generated and analyzed during the current study are available from the corresponding author (F.K.) upon reasonable request.

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

### Acknowledgements
This work was supported by the Army Research Office, the Innovation Fund Denmark, the Villum Foundation, the Danish National Research Foundation, and the ARC via the Centre of Excellence in Engineered Quantum Systems (EQuS), project number CE170100009. Work at Purdue was supported by the U.S. Department of Energy, Office of Basic Energy Sciences, Division of Materials Sciences and Engineering under Award No. DE-SC0006671. Additional support from Nokia Bell Labs for the GaAs MBE effort is also gratefully acknowledged.

### Author contributions
S.F., G.C.G. and M.J.M. developed the GaAs heterostructures. P.D.N. fabricated the devices. F.K.M. and F.M. carried out the measurements with input from C.M.M. and F.K. Data analysis was done by F.K.M. and F.M., and theoretical models and simulations were developed by T.B.S., S.D.B., and A.C.D. All authors contributed to interpreting the data. The manuscript was written by F.K.M., F.M. and F.K. with suggestions from all other authors.

### Additional information

**Competing interests:** The authors declare no competing interests.

