## [Peer Review File · Nature Communications]

Reviewers' comments:

Reviewer #1 (Remarks to the Author):

The Author's replies to the points raised by the Referees are detailed and pertinent. I am satisfied with the replies and the revised manuscript which I recommend for publication in Nature Communications.

Reviewer #2 (Remarks to the Author):

The authors gave a detailed answer to all the points raised in my first report and the report of the second referee. Most of the answers were quite useful to clarify the manuscript and it resulted in modifications of the manuscript. Nevertheless, on the two critical points I raised in my report, it was surprising that these were not addressed accordingly the modification.

On the novelty side, the authors gave a complete list of the novel achievements that are present in their paper. All of them were clear to me when I wrote my first report. They are specific to the field of spin qubits and for specialists to appreciate. It does not really change the big picture and my comment on the novelty. I reassert here that the quality of the data and the depth of the analysis are noteworthy. Nevertheless, the existence of the Delft paper and its relation to the present work deserve a fair and comprehensive comparison in the introduction, and in my opinion, it should be included based on the author's response.

On the tunability side, I am not convinced by the answer of the authors. Not showing a more complete stability diagram of the 5 dot structure is problematic. The level of complexity of a five dot structure and its related difficulty to tune is well acknowledged within the community. Let me assert that it is far from a routine characterization procedure nowadays and it is quite different to what they reported so far on only 3 dot structure in the two papers that they mentioned in the answer. The data presented in FigS3 are not so clear to me and it did reflect the difficulty to have a proof of the charge configuration of the many quantum dot systems. I think it is important to know the number of electrons in the different dots. It can change the interpretation of the physical process and on the strength of the coupling. I need more data or more details of the procedure to be convinced of the charge configuration of the five dot system claimed in the manuscript.

I have one last question on the manuscript: The authors claim in their response that the physics mechanism at play is different to the one of the Delft paper and it is reminiscent of the RKKY interaction. According to me, it is not demonstrated experimentally in the paper (and it is not claimed as well). Am I correct? Can the authors provide insights on how to distinguish these two processes?

December 4, 2018

We thank both referees for taking the time to review our revised manuscript for Nature Communications. Referee 1 is satisfied with our revisions and recommends publication. Referee 2 found our revisions useful and recommends that our clarification of two critical points are incorporated in the manuscript. We have followed this recommendation (see below). As requested by the editor, in the revised manuscript, changes are shown in blue.

(1) Relation of our work to earlier work reported by Delft [17].

We have added a brief comparison between our observations and Delft's work in the "Discussions" section, provide a more detailed discussion in Supplementary Section VIII, and are citing Delft's work in the introductory paragraph. These additions are based on our response to the referee, but shorter due to overall length considerations. In addition to this narrow comparison, we have also added a broader comparison with resonator-based approaches to coupling spin qubits. This was also included in our earlier response to the referee, and given the recent "breakthrough" publications that appeared in Nature and Science (after our data was analyzed) we felt that this comparison is indeed appropriate. We also comment on RKKY interactions (see below).

(2) More complete stability diagrams of the 5-dot structure.

The referee found the absence of stability diagrams problematic (due to the complexity of a five dot structure and the absence of standard procedures in literature) and correctly states that it is important to know the number of electrons in the different dots. The presence of two radio-frequency charge sensors indeed allowed us to measure (two-dimensional) cross-sections of the high-dimensional (more than 10 relevant tuning voltages) stability diagram associated with the array. Each double-dot occupation, (2,0), (1,1), or (0,2), was established using standard charge sensing procedures with high confidence, and is detailed in Section II of Ref. [4]. As stated on p. 1, the absolute occupation of the multi-electron dot is unknown (and the manuscript provides an estimate), but its occupation can be changed controllably by adding or removing one or several electrons (as shown in Fig. 1c in Ref [4]). This allowed us to choose an even occupation number for this experiment. This freedom of choice is not new [4], and hence our manuscript was brief on this procedure. For clarity, we have added an additional reference to Ref. [4] on p. 1: *Its ground state is chosen to be spinless as described in Ref. [4]. Two two-electron double dots are tunnel-coupled [4] on opposing sides*

of the large middle dot and are each initialized and read out using standard techniques for singlet-triplet qubits [5,25].

Finally, the referee is curious whether we can make a firm statement about differences between physical processes in our device and those in the Delft device. Unfortunately, the data available in [17] is somewhat inconclusive, making this rather difficult. In our understanding, the prominent oscillations in [17] are conventional direct exchange, induced by electrostatically lowering the unoccupied (1,1,0) state towards an asymmetric charge configuration, namely towards the charge transition between (2,0,0) and (1,0,1) (cf. Fig. 1c and Fig. S6 of [17]). These direct exchange processes (Fig. 3a and Fig. S6 in [17]) can be contrasted to exchange processes between a spin-full multielectron dot and a spin-full one-electron dot, which show phenomena that are absent and likely impossible in one-electron dots (negative exchange, electron-electron correlations, see ref [4]). Delft also observed fainter oscillations just before the onset of direct exchange oscillations (but only in the highly asymmetric electrostatic configuration indicated in Fig. 1c), and claimed that these arise from what they call “superexchange”. In their definition, the two characteristic features of this type of exchange is a negative sign and a dependence on tunnel coupling to the fourth power. None of these defining characteristics is experimentally supported. In fact, from Fig. 3a we understand that the observed “superexchange” oscillations have the same sign as the direct exchange (which is also visible in the upper half of Fig. 3a). Due to these differences, we do not wish to speculate firmly about the Delft mechanism. Concerning the mechanism underlying our experiment, we believe that ref [19] gives some insight, and therefore we have included in our revised discussion section a statement about the possible role of RKKY interactions. An experimental demonstration of RKKY may be possible by studying larger dots (which we proposed in the outlook section), although one would need to carefully work out how the distance dependence interplays with the distance-dependent charging energy. Another experimental test would involve quantifying the tunnel matrix elements, and studying the t^4 dependence in the on state, and the charging-energy dependence in the off state. A characteristic feature of RKKY interaction is the emergence of interference effects between different tunneling paths (which requires different orbitals), leading to oscillations and sign changes in the mediated spin exchange. In our understanding of ref [19], the smallest system that can be considered reminiscent of RKKY involves only one (fully occupied) orbital, which would not be rich enough to show oscillations due to interference effects. Again, larger dots may be an interesting approach to clarifying the presence of such interference effects.

REVIEWERS' COMMENTS:

Reviewer #2 (Remarks to the Author):

The authors have now answered one of my questions in a satisfactory manner, concerning the precise comparison to the Delft publication and I think the discussion improves the quality of the manuscript. The second question concerning the charge configuration is still not appropriately addressed and my comment in the previous report did not have much impact on the answer of the authors and on the manuscript. Considering the quality of the data presented in the manuscript, I want to be sure that it is not a simple problem of communication and therefore I propose to try again to clarify my request to the authors. I am clearly aware of the publication of the group and their ability to initialize double or triple dot system with one electron in each dot. From their answer, I understand that the authors did that for the two double dots on the side of the five-dot system, before bringing all the dots in interaction in the so-called "spin interaction point". To reach this point, I expect that a change in gate voltage configuration is needed and such a change can cause changes in the charge configuration. Usually a new stability diagram at or close to the spin-interaction point is needed to check the charge configuration. As I understand the manuscript, this is exactly the purpose of FigS3 in the manuscript. The problem is that I have difficulty to draw any conclusions about the charge configuration from the figure S3 due to the poor quality of the presented data and I am not convinced by the interpretation of the authors concerning this particular figure. As a consequence, I would recommend that the authors give additional details and data regarding their procedure to assign the charge configuration or to lower their claim around the charge configuration and explaining clearly their procedure and how they estimated the charge occupation of the dot system.

February 2, 2019

We thank Reviewer #2 for taking the time to clarify their question about identification of charge configurations.

Reviewer #2: The authors have now answered one of my questions in a satisfactory manner, concerning the precise comparison to the Delft publication and I think the discussion improves the quality of the manuscript. The second question concerning the charge configuration is still not appropriately addressed and my comment in the previous report did not have much impact on the answer of the authors and on the manuscript. Considering the quality of the data presented in the manuscript, I want to be sure that it is not a simple problem of communication and therefore I propose to try again to clarify my request to the authors. I am clearly aware of the publication of the group and their ability to initialize double or triple dot system with one electron in each dot. From their answer, I understand that the authors did that for the two double dots on the side of the five-dot system, before bringing all the dots in interaction in the so-called “spin interaction point”. To reach this point, I expect that a change in gate voltage configuration is needed and such a change can cause changes in the charge configuration. Usually a new stability diagram at or close to the spin-interaction point is needed to check the charge configuration. As I understand the manuscript, this is exactly the purpose of FigS3 in the manuscript. The problem is that I have difficulty to draw any conclusions about the charge configuration from the figure S3 due to the poor quality of the presented data and I am not convinced by the interpretation of the authors concerning this particular figure. As a consequence, I would recommend that the authors give additional details and data regarding their procedure to assign the charge configuration or to lower their claim around the charge configuration and explaining clearly their procedure and how they estimated the charge occupation of the dot system.

We believe there was a small misunderstanding earlier, and the referee’s clarification of the question helped us provide a better answer: The purpose of Fig. S3 (now Suppl. Fig. 4) is *not* to support the charge occupation of the double dots once the 5-dot system is tuned up and ready for application of exchange pulses. The purpose of Fig. S3 is to provide evidence for the charge occupation of the inner dots and multielectron dot *during* the exchange pulse, and how it depends on pulse amplitude and detuning. The charge state of each double dot *right before* application of an exchange pulse can be verified using conventional techniques, using the charge sensor signal as a function of the two plunger voltages within each double dot. The charge state of the

middle dot *during* the exchange step is more difficult to measure directly, as the exchange step is only few nanoseconds long, too short to get reliable data from the RF charge sensors.

Concerning the charge distribution, there are therefore two different confidence levels for the charge identification in the two stages in our experiments:

(1) Charge state configuration of the five-dot array tuned up ready for application of exchange pulses

At this stage we are very certain about the charge configuration. That's because we can make use of the two charge sensors to determine the charge occupation of each double dot individually (even in the presence of the multi-electron dot). In the figure below we show two charge diagrams where the charge configuration ((2,0), (1,1), or (0,2)) can be clearly identified for both the left double dot and the right double dot. This data was taken with the middle multi-electron dot already tuned up, using techniques presented in Section II of Ref. 4.

We have clarified this issue by revising Supplementary Note I as follows (changes shown in bold):

For each double dot (i.e. left and right double dot separately) we establish a partial charge stability diagram, by sweeping its plunger gates ($V_{L1,L2}$ or $V_{R1,R2}$) while monitoring its charge sensor. **This can be done despite the presence of the multi-electron dot, provided that plunger gates are swept sufficiently slow to allow exchange of electrons between the double dots and their reservoirs.** (Compared to traditional double dots, each double dot only has one reservoir just to the left (right) of the left (right) double dot shown in Figure 1a, which may require cotunneling processes in order to transition into the charge ground state of the inner quantum dots.) Application of (unoptimized) pulse

sequences (corresponding to double-dot leakage-spectroscopy measurement at fixed, finite magnetic field) allows us to optimize the static gate voltages associated with each double dot ...

(2) Charge state of the system during the exchange step

The total charge of the five-dot system during the exchange step can be assumed to be the same as just before arrival of the exchange pulse, if (i) an exchange of electrons with the reservoirs is prevented by sufficiently high barriers to the reservoirs [Supplementary Reference 5] or (ii) if positive voltage pulses applied to the multielectron dot are compensated for by appropriate negative compensation pulses on the neighboring plunger gates of the double dots (such that the overall charge occupation electrostatically remains the ground states, see Supplementary Note IV).

Therefore, we think that the disputed question here is not about the total charge of the 5-dot array, but about the charge distribution within the 5-dot array (during the exchange pulse). The charge state during the exchange pulse is, indeed, more difficult to measure, as the nanosecond-scale pulse duration is too short to give meaningful data from the two charge sensors. (The time response of each charge sensor is approximately 10ns, limited by the quality factor of the rf tank circuit.) Currently our best way to read the sensors during the exchange pulses is to temporarily increase their duration to 4 μ s, while keeping all other pulse parameters unchanged. Due to the capacitive coupling between the gate electrode V_M and the sensors, this means that the operating point of the sensor "falls off" the sensitive Colom peak when parameters ε and ε_M are changed. As explained in the discussion of Suppl. Fig. 4, we have therefore acquired this ε - ε_M map for various different settings of the sensor dot, and averaged these maps as described in detail in Supplementary Section IV. This explains why Suppl. Fig. 4 is more difficult to interpret.

The dashed lines in this figure represent our interpretation in terms of charge transitions. Our interpretation of Suppl. Fig. 4 is corroborated by our simulations in Figs 3 (e) and (f) as well as recent work in Supplementary Reference 5 where arrays of quantum dots are operated in metastable states in an overall isolated regime. Consequently, we are certain about the inferred charge distribution and wish to maintain our claims concerning the occupation of the quantum dots. Following the referee's suggestion to be more clear about the nature of our claims, we have improved the presentation in Supplementary Note IV as follows (additions in bold):

In an attempt to independently confirm the position of the electrons during

the interaction step, we extend the interaction time to $4 \mu\text{s}$ while keeping the remainder of the pulse sequence unchanged [...]. **Our interpretation in terms of charge transitions within the five-dot array are indicated with dashed black lines. [...]. Our interpretation of Suppl. Fig. 4 is corroborated by our simulations in Figs. 3e and 3f as well as recent work in Supplementary Reference 5, where arrays of quantum dots are operated in metastable states in an overall isolated regime.**

We hope that this clarification answers the referee's question.